# Diagnosis of Progressive Disseminated Histoplasmosis in Advanced HIV: A Meta-Analysis of Assay Analytical Performance

**DOI:** 10.3390/jof5030076

**Published:** 2019-08-18

**Authors:** Diego H. Caceres, Martha Knuth, Gordana Derado, Mark D. Lindsley

**Affiliations:** 1Centers for Disease Control and Prevention, Mycotic Diseases Branch. Atlanta, GA 30333, USA; 2Studies in Translational Microbiology and Emerging Diseases (MICROS) Research Group, School of Medicine and Health Sciences, Universidad del Rosario, Bogota 11011, Colombia; 3Centers for Disease Control and Prevention, Center for Surveillance, Epidemiology, and Laboratory Services (CSELS), Division of Public Health Information Dissemination (DPHID), Atlanta, GA 30333, USA

**Keywords:** histoplasma, histoplasmosis, HIV, culture, serology, antibody, antigen, PCR, molecular assays, diagnosis, analytical performance

## Abstract

Histoplasmosis is an important cause of mortality in people with advanced HIV, especially in countries with limited access to diagnostic assays. Histoplasmosis can be diagnosed using culture, histopathology, and antibody, antigen, and molecular assays. Several factors may affect the analytical performance of these laboratory assays, including sample type, clinical stage of the disease, and previous use of antifungal treatment, among others. Here we describe the results of a systematic literature review, followed by a meta-analysis of the analytical performances of the diagnostic laboratory assays employed. Our initial search identified 1631 references, of which 1559 references were excluded after title and abstract screening, leaving 72 references identified as studies relevant to the validation of histoplasmosis diagnostic assays. After evaluating the full text, 30 studies were selected for final review, including one paper not identified in the initial search. The meta-analysis for assay analytical performance shows the following results for the overall sensitivity (Sen) and specificity (Spe) of the various methods evaluated: Culture, Sen 77% (no data for specificity calculation); antibody detection assays, Sen 58%/Spe 100%; antigen detection assays, Sen 95%/Spe 97%; and DNA detection assays (molecular), Sen 95%/Spe 99%. Of the 30 studies reviewed, nearly half (*n* = 13) evaluated *Histoplasma* antigen assays, which were determined to be the most accurate methodology for diagnosis of progressive disseminated histoplasmosis in advanced HIV (inverse of the negative likelihood ratio was 13.2). Molecular assays appear promising for accurate diagnosis of histoplasmosis, but consensus on exact techniques is needed. Cultures showed variable sensitivity related to sample type and laboratory handling. Finally, antibody assays presented high specificity but low sensitivity. This poor sensitivity is most likely due the highly immunosuppressed state of this patient population. Diagnostic assays are crucial for accurate diagnosis of progressive disseminated histoplasmosis (PDH) with advanced HIV disease.

## 1. Introduction

Histoplasmosis is a disease caused by the thermally dimorphic fungus *Histoplasma capsulatum*. This disease has been reported worldwide, but is most frequently diagnosed in the Americas [1]. *H. capsulatum* is frequently found in soil, especially where it is contaminated with bird excreta and bat guano [2]. *H. capsulatum* primarily causes pulmonary infection when the human host inhales infectious propagules (microconidia and mycelial fragments) after soil disturbance. It can spread secondarily to other organs, especially those of the reticuloendothelial system [2]. In persons with advanced Human Immunodeficiency Virus (HIV), infection often develops into a clinical form called progressive disseminated histoplasmosis (PDH), where the fungus disseminates to other parts of the body, resulting in high mortality if not treated early [2,3,4]. PDH symptoms are nonspecific, and among people living with HIV (PLHIV), the symptoms may be similar to those of other infectious diseases, in particular to tuberculosis (TB), thus complicating diagnosis and treatment [5,6,7].

The gold standard for diagnosis of histoplasmosis is based on conventional laboratory assays using culture and histopathology (including special stains) [8]. These assays have several limitations, including the need for high-level laboratory infrastructure for culture handling (biosecurity level 3) the need for highly trained laboratory staff, variable assay analytical performance, and a long turn-around time for results [9,10]. Other alternatives for histoplasmosis diagnosis include assays for the detection of specific host antibodies against *Histoplasma* antigens; detection of circulating *Histoplasma* antigens in urine, serum, and bronchoalveolar lavage (BAL); and detection of fungal DNA [11].

The analytical performance of the assays for the diagnosis of histoplasmosis varies according to disease stage and clinical form. For that reason, the aim of our study was to perform a systematic review of the literature and a meta-analysis to evaluate the analytical performance of laboratory assays for the diagnosis of PDH in PLHIV.

## 2. Materials and Methods

### 2.1. Literature Search

We searched the following databases on 20 February 2019 for the terms histoplasmosis, HIV, and terms for diagnostics assays evaluated, including their synonyms, in the title, abstract, keywords, or subject headings: Medline (Ovid), Embase (Ovid), CAB Abstracts (Ovid), Global Health (Ovid), Scopus, the Cochrane Library, PubMed Central, and LILACS. We also conducted a broader search on 20 February 2019 in the same databases for histoplasmosis, HIV, and a diagnostic methodology search filter adapted from the McMaster Health Information Research Unit’s recommended search hedges [12]. These searches were limited to those studies published in English, Spanish, and Portuguese. Complete search strategies for each database are given in the Appendix A.

### 2.2. Study Selection Criteria

Studies were included in the analysis if they demonstrated validation of *Histoplasma* laboratory assays. Studies were excluded if they were not focused on human application or were primarily case reports, clinical studies, environmental or epidemiological studies, or literature reviews with no validation component. For studies related to validation of laboratory assay for the diagnosis of histoplasmosis, we excluded studies performed on patients without HIV, concordance studies, and studies without a clear number of patients tested. To maintain the accuracy of the study, references were not included in the analysis if culture or histopathological analysis were not included to determine proven cases as defined by the EORTC/MSG Consensus Group [8]. This report was done using the PRISMA statement [13].

### 2.3. Statistical Analysis and Data Synthesis

Analysis was performed using STATA’s *metandi* and *metan* commands [14].

### 2.4. Calculation of Assays Analytical Performance

The number of patients classified as true positive (TP), false negative (FN), false positive (FP) and true negative (TN) in the results were extracted from selected studies (Appendix A). Using these data, 2 × 2 tables were constructed to estimate each assay’s sensitivity, specificity, positive and negative likelihood ratios (LR+ and LR–, respectively) and their 95% confidence intervals (95%CI). For the evaluation of the assay’s analytical performance, the following factors were considered: (A) Excellent analytical performance: LR- <0.1 and LR+ >10; (B) Good analytical performance: LR– 0.1–0.2 and LR+ 5–10; (C) Low analytical performance: LR– 0.21–0.5 and LR+ 2–4.9; and (D) Poor analytical performance: LR– 0.51–1 and LR+ <2 [14,15].

### 2.5. Meta-Analysis Forest Plot

The analytical performance of the assays and meta-analysis results were graphically displayed using forest plots. Graphics represent summary measures and confidence intervals, and squares are proportional to the weights (study sample size) used in the meta-analysis [14].

### 2.6. Plot of Fitted Model

The resulting graph shows the following summaries, along with circles showing the individual study estimates: (i) a summary curve from the hierarchical summary receiver operating characteristic (HSROC) model; (ii) a summary of operating points (summary values for sensitivity and specificity); (iii) a 95% confidence region for the summary operating point; (iv) and a 95% prediction region (the confidence region for a forecast of the true sensitivity and specificity in a future study) [14]. Pairs of sensitivity and specificity extracted from each primary study are plotted on a ROC space, showing between-study heterogeneity as well as a relationship between sensitivity and specificity. The HSROC curve is plotted as a curvilinear line passing through summary point [14].

### 2.7. Hierarchical Summary Receiver Operating Characteristic (HSROC) Curves

HSROCs were used to evaluate the assays’ accuracy. The positive and negative likelihood ratios (LR+ and LR–) were estimated. Larger values of the LR+ and of 1/LR– both indicate higher accuracy of the assay and can be compared to evaluate whether a positive or negative test result has a greater impact on the odds of disease. HSROC confidence regions and prediction regions were used to evaluate heterogeneity within and between studies [14].

## 3. Results

### 3.1. Literature Search

We identified 1259 references using the narrow search and 2228 references with the broad search. After the removal of duplicate references, 1631 references were evaluated. A total of 1559 references were excluded after a manual review of titles and abstracts; the reasons for exclusion are summarized in Figure 1. Seventy-two studies were related to the validation of diagnostic assays for histoplasmosis in PLHIV. Of the 72 publications, 29 were selected; seven related to the validation of culture methodologies, five were related to assays for detection of antibodies, twelve were related to detection of circulating antigen, and five were related to molecular testing. By expert opinion, one study related to antigen testing validation that was not identified in the systematic review was included, increasing the number of studies related to antigen assay validation to thirteen and the total number of publications selected to 30, Figure 1 [16,17,18,19,20,21,22,23,24,25,26,27,28,29,30,31,32,33,34,35,36,37,38,39,40,41,42,43]. There were no identified studies related to histopathology validation.

### 3.2. Meta-Analysis

#### 3.2.1. Evaluation of Culture Assays

Seven studies were identified [16,17,18,19,20,21,22]. Of that, two had enough available information to calculate the analytical performance of the culture according to the type of sample [21,22]. The overall sensitivity for culture was 77% (95%CI 72%–81%), Figure 2. Studies that evaluated blood culture processed by lysis methodology and those using bone marrow, Figure 2 (studies 1, 2 and 4) showed the highest analytical performance; sensitivity ranged between 60% and 90%. Respiratory samples presented poor sensitivity (from 0% to 60%). None of these studies have a non-histoplasmosis control group; for that reason, it was not possible to calculate assay specificity or HSROC curve.

#### 3.2.2. Evaluation of Antibody Detection Assays

Five studies were identified in this category. These studies evaluated different methodologies for antibody detection, including Western blot (WB), immunodiffusion (ID), complement fixation (CF), enzyme-linked immunosorbent assay (ELISA), and counter immune-electrophoresis (CIE) [23,24,25,26,27]. For antibody detection assays, the overall sensitivity was 58% (95%CI 53%–62%), and the overall specificity was 100% (95%CI 99%–100%), Figure 3. WB and ELISA methods showed the highest analytical performance; Figure 3 (studies 1 and 5) with sensitivity of 90% and 86%, respectively). All the antibody detection studies reported here demonstrated high specificity (greater than 90%). Antibody detection assays presented low analytical performance (1/LR– = 2.3 [95%CI 1.4–3.7], LR+ = 1146.3 [95%CI 8.5–154,326.2]), indicating the least accuracy diagnosing histoplasmosis in PLHIV, Table 1.

#### 3.2.3. Evaluation of Antigen Detection Assays

This methodology had the most studies included (*n* = 13 studies) [24,26,28,29,30,31,32,33,34,35,36,37,38]. Most of the reports evaluated ELISA, and one study described the validation of point-of-care (POC) testing, which provided results in less than an hour [29]. Detection of circulating *Histoplasma* antigens showed an overall sensitivity of 95% (95%CI 94%–97%) and an overall specificity of 97% (95%CI 97%–98%), Figure 4. Antigen assays presented a 1/LR– of 13.2 (95%CI 7.7–22.7), and a LR+ of 18.7 (95%CI 11.7–30.1), indicating an excellent analytical performance. These types of assays were the most accurate at diagnosing histoplasmosis in PLHIV (Table 1). These studies evaluated five different antibody preparations for the detection of antigen, three polyclonal antibodies [24,26,29,30,31,32,33,34,36,37,38], and two monoclonal antibodies [28,35].

#### 3.2.4. Evaluation of Molecular Assays

Five studies were identified for molecular testing. These methodologies evaluated different types of specimens, including respiratory, tissue biopsy, blood and bone marrow samples [39,40,41,42,43]. There was no consensus in PCR protocols and gene targets (amplification of internal transcribed spacer [ITS] regions and a 100-kDa-like-protein genes), and all PCR protocols were in-house protocols. For assays based on the detection of *Histoplasma* DNA, the overall sensitivity was 95% (95%CI 89%–100%) and overall specificity was 99% (95%CI 96%–100%) (Figure 5). Similar to the antigen testing, molecular assays presented excellent analytical performance (LR+ = 70.7 [95%CI 7.2–691.9], and 1/LR- = 12.3 [95%CI 3.6–41.1]), displaying a similar accuracy to the assays based on antigen detection (Table 1).

## 4. Discussion

Using a literature search and meta-analysis, we evaluated the analytical performance of multiple laboratory assays used to diagnose progressive disseminated histoplasmosis in PLHIV with advanced HIV disease. The results of the meta-analysis showed that laboratory assays based on the detection of circulating *Histoplasma* antigen demonstrated the best analytical performance. However, these results may be partially related to the number of antigen assay studies that were available for evaluation (*n* = 13) compared to those available to evaluate the other methodologies (*n* = 5–7).

Because validated *Histoplasma* antigen assays are commercially available, incorporating this testing into clinical laboratories is often easier, reducing the technical problems that can be related with performance of in-house assays. In locations where these assays have been implemented, the performance of these more sensitive and specific assays have resulted in a significantly increased number of patients diagnosed compared with conventional diagnostic tests (culture and histopathology) [28,30,31]. Likewise, the relative ease and speed in which these antigen assays can be performed has resulted in a shortened time of diagnosis and a reduction of mortality associated with PDH in PLHIV [44,45,46,47]. Antigen assays can also be performed in laboratories that do not have the higher levels of biocontainment needed for handling *Histoplasma* cultures for conventional identification or DNA extractions for molecular identification. An advantage of ELISA-based antigen assays is that they are designed to easily evaluate a larger number of patient specimens. However, a limitation is that some of these assays require multiple wells (up to 8–10 wells) for quality control and standard curve formation, reducing the cost effectiveness of the assaying when only a small number of samples are tested. Development of simple and rapid diagnostic technology, like laboratory assays based on immunochromatographic assays are being explored [29].

The results from this study also demonstrated that molecular testing may be another alternative for accurate diagnosis of histoplasmosis in PLHIV. However, the lack of commercially available kits, the lack of standardized methods, and the limited number of validation studies are current limitations in performing these assays in laboratories. It has yet to be determined which DNA extraction method, gene target and primers, or amplification methodology is optimal, which will most likely vary between the use in culture confirmation and the direct detection in patient specimens. Furthermore, the definitions of invasive fungal diseases published by the EORTC/MSG Consensus Group in 2008 do not include molecular assay as a methodology for the diagnosis of endemic mycoses due to a lack of consensus on gene targets and laboratory protocols for molecular assays. Further investigations, laboratory consensus and development of commercial kits are needed [8,48].

Culture is still considered the gold standard for the diagnosis of histoplasmosis, however the results of this literature review and meta-analysis show variable sensitivity when the cultures are not properly processed (i.e., blood culture processed by lysis centrifugation versus not processes by lysis centrifugation). Selection of the proper specimen is crucial; blood and bone marrow cultures provided the highest assay sensitivity, while the other specimen types demonstrated lower sensitivity. Additionally, there are several limitations in performing the culture for the diagnosis of histoplasmosis. Primarily, the need for laboratory infrastructure for handling isolates (biosafety level 3), the need for staff who have appropriate laboratory training and experience, and a prolonged turn-around time for results. This final variable is directly associated with an increased risk of patient mortality [49]. However, conventional assays like culture or special histopathologic stains may be the only option for laboratory diagnosis of certain forms of histoplasmosis (e.g., muco-cutaneous, central nervous system and pulmonary localizations) [3].

Results from this meta-analysis demonstrated that antibody detection assays have high specificity, but the sensitivity of this type of diagnostic assay in PLHIV with advanced diseases was poor. These findings can be explained by the highly immunosuppressed state of this patient population. However, the detection of specific anti-*Histoplasma* antibodies in patient’s sera or cerebrospinal fluid could be a complementary diagnostic tool, particularly in PLHIV with subacute and chronic forms of PDH where patients have a progressive long-term infection characterized by a lower fungal burden [3,11]. Additionally, it is important to mention that antibody detection assays are useful for the diagnosis of histoplasmosis in non-immunosuppressed patients (subacute and chronic histoplasmosis) [11].

## 5. Conclusions

Diagnostic assays are crucial to improving the care of patients with advanced HIV disease who are most at risk of developing progressive disseminated histoplasmosis (PDH). Results from this meta-analysis demonstrated that antigen and molecular diagnostic assays had greater sensitivity and specificity in identifying PDH in PLHIV compared with culture and antibody assays. In contrast, multiple diagnostic assays may be helpful in identifying non-disseminated histoplasmosis. Further investigation, diagnostics developments, and guidelines for diagnosis are needed to improve capacity to rapidly detect PDH in PLHIV. Finally, this systematic review is limited to the analysis assays performance on PDH in PLHIV, due the variety of histoplasmosis clinical presentations, further systematic reviews and meta-analysis are needed.

## Figures and Tables

**Figure 1 jof-05-00076-f001:**
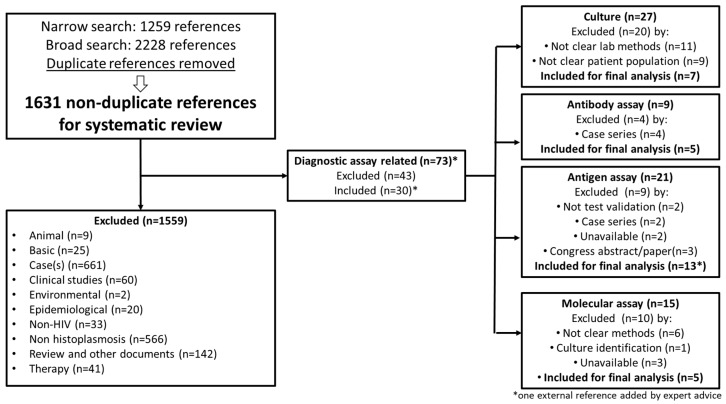
A flow chart of the literature search and studies selection.

**Figure 2 jof-05-00076-f002:**
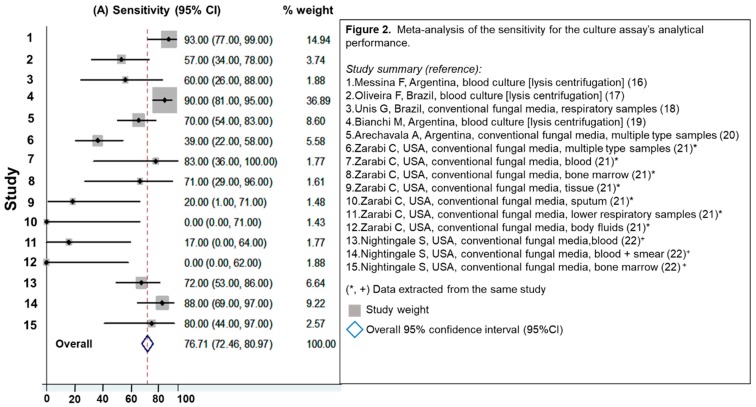
Meta-analysis of the sensitivity for the culture assay’s analytical performance.

**Figure 3 jof-05-00076-f003:**
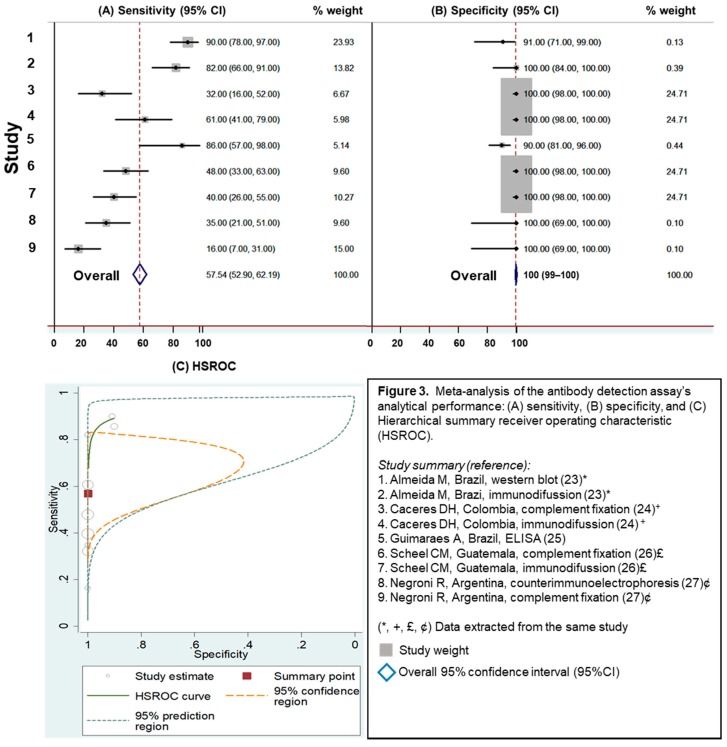
Meta-analysis of the antibody detection assay’s analytical performance: (**A**) sensitivity, (**B**) specificity, and (**C**) Hierarchical summary receiver operating characteristic (HSROC).

**Figure 4 jof-05-00076-f004:**
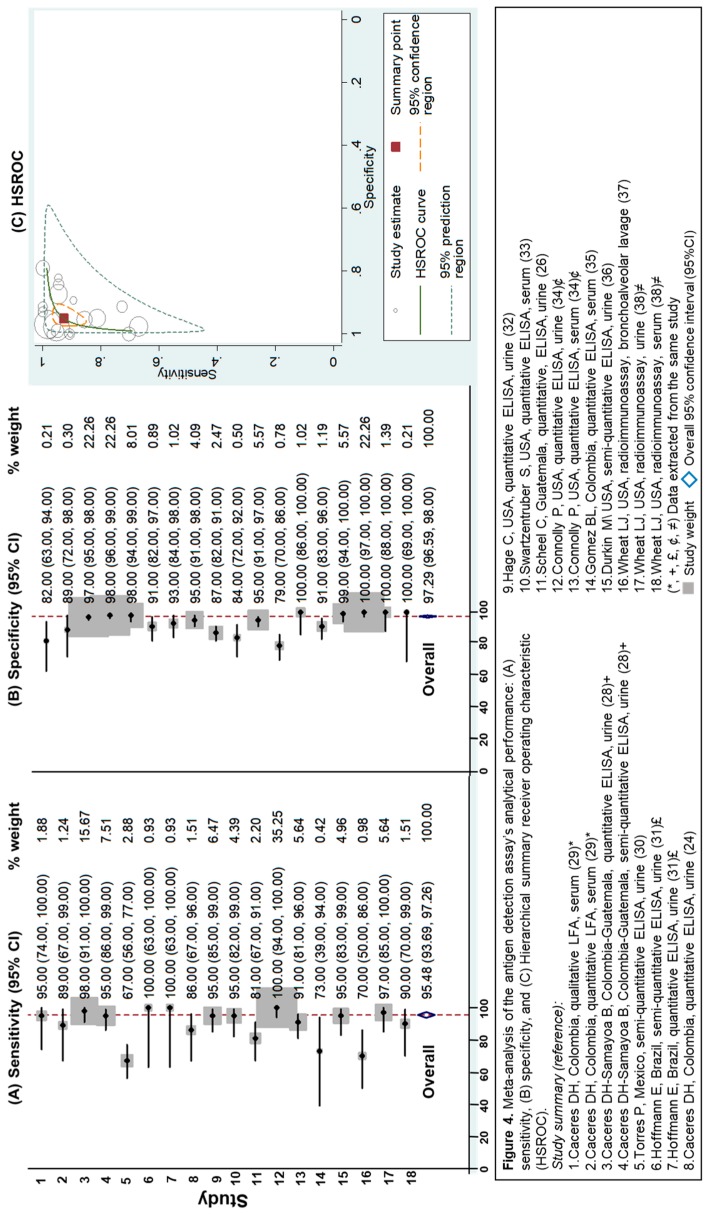
Meta-analysis of the antigen detection assay’s analytical performance: (**A**) sensitivity, (**B**) specificity, and (**C**) Hierarchical summary receiver operating characteristic (HSROC).

**Figure 5 jof-05-00076-f005:**
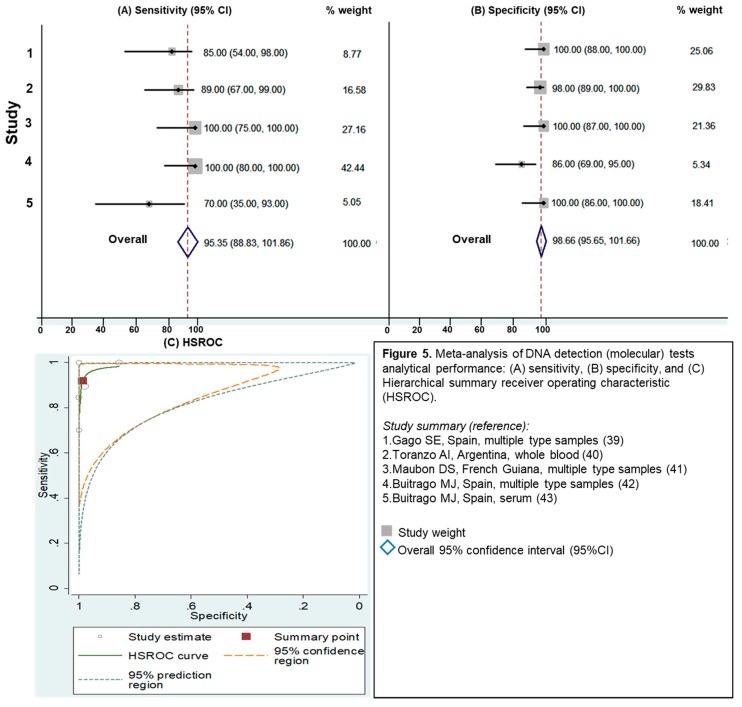
Meta-analysis of DNA detection (molecular) tests analytical performance: (**A**) sensitivity, (**B**) specificity, and (**C**) Hierarchical summary receiver operating characteristic (HSROC).

**Table 1 jof-05-00076-t001:** Summary of the meta-analysis of the analytical performance of the assays for the diagnosis of PDH in PLHIV.

Assay	Sensitivity% (95% CI)	Specificity% (95% CI)	Accuracy% (95% CI)	LR+/LR–	1/LR– (95% CI)
**Antibody detection assays**	58 (53–62)	100 (99–100)	89 (87–91)	1146.3 (8.5–154326.2)/0.4 (0.2–0.7)	2.3 (1.4–3.7)
**Antigen detection assays**	95 (94–97)	97 (97–98)	95 (94–96)	18.7 (11.7–30.1)/0.07 (0.04–0.13)	13.2 (7.7–22.7)
**Molecular assays**	95 (89–100)	99 (96–100)	96 (94–99)	70.7 (7.2–691.9)/0.08 (0.02–0.27)	12.3 (3.6–41.1)

(LR–) Negative likelihoods, (LR+) Positive likelihoods, (1/LR) Inverse of the negative likelihood ratio; (95%CI) 95% confidence interval.

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
