# Peer review of "Diagnosis of Progressive Disseminated Histoplasmosis in Advanced HIV: A Meta-Analysis of Assay Analytical Performance"

_jof, 2019, doi:10.3390/jof5030076_

Round 1

Reviewer 1 Report

This is a metanalysis of studies of diagnostics tests for disseminated histoplasmosis in AIDS patients.  The study is well done but the results are not unexpected and of questionable novelty. 

The data are presented in far too much detail - there are 6 figures and one summary table, as well as data in the text.  This could be condensed into a much smaller number of Figures and/or tables.  The HSROC curves don't add anything, in my openion, and the reference you make to explain how the test is done is not easily available.

One thing that would have made this paper more interesting is a discussion of the differences between the usefulness of antibody tests in AIDS vs other patient populations and/or a discussion of molecular methods and optimal methods of culture.  There is a problem with reference 29.

Author Response

Reviewer 1:

This is a metanalysis of studies of diagnostics tests for disseminated histoplasmosis in AIDS patients. The study is well done but the results are not unexpected and of questionable novelty.

The data are presented in far too much detail - there are 6 figures and one summary table, as well as data in the text. This could be condensed into a much smaller number of Figures and/or tables. The HSROC curves don't add anything, in my openion, and the reference you make to explain how the test is done is not easily available.

R. Thanks for your comments. We would like to keep the presentation of the results in the way the manuscript was submitted. Data was prepared and presented based on the recommendation of two CDC’s statisticians (Gordana Derado and Anna J. Blackstock)

One thing that would have made this paper more interesting is a discussion of the differences between the usefulness of antibody tests in AIDS vs other patient populations and/or a discussion of molecular methods and optimal methods of culture.

R. This systematic review and meta-analysis was done in order to support the development of the WHO guidelines for Guidelines on the Diagnosis and Management of Progressive Disseminated Histoplasmosis in People living with HIV; that is the reason why the focus population is people with advanced HIV disease. In order to clarify it, we modified manuscript title as follow: “Diagnosis of Progressive Disseminated Histoplasmosis in Advanced HIV: a Meta-analysis of Assay Analytical Performance”, and additionally, we described study limitations in the conclusion section as follows: “Finally, this systematic review is limited to the analysis assays performance on PDH in PLHIV, due the variety of histoplasmosis clinical presentations, further systematic reviews and meta-analysis are needed”.

Based in your recommendation, we clarified that antibody detection testing is useful on the diagnosis of other form of histoplasmosis (sub-acute and chronic disease) on page 12, line 225. Data available on molecular methods and culture testing is limited and not well supported.

There is a problem with reference 29.

R. Thanks, we fix it.

Reviewer 2 Report

The manuscript by Caceres et al. describes a meta-analysis of published information related to the diagnosis of histoplasmosis in HIV patients. The findings are interesting and support the idea that serological assays are the best choice for diagnosis of the disease. I have some points though to improve this study:

Although I agree with the strategy used by the authors for literature search, I think this should include all the studies related to histoplasmosis diagnosis, regardless of the co-infection with HIV. It would be of special interest to assess whether the calculated figures change or not when including these cases.

The section dealing with the molecular diagnosis could be improved including a brief revision of the sequences amplified, amplicon sizes, etc.

The addresses of the websites used for literature search should be included in the manuscript.

Author Response

Reviewer 2:

The manuscript by Caceres et al. describes a meta-analysis of published information related to the diagnosis of histoplasmosis in HIV patients. The findings are interesting and support the idea that serological assays are the best choice for diagnosis of the disease. I have some points though to improve this study:

Although I agree with the strategy used by the authors for literature search, I think this should include all the studies related to histoplasmosis diagnosis, regardless of the co-infection with HIV. It would be of special interest to assess whether the calculated figures change or not when including these cases.

R. This systematic review and meta-analysis was done in order to support the development of the WHO guidelines for Guidelines on the Diagnosis and Management of Progressive Disseminated Histoplasmosis in People living with HIV; that is the reason why the focus population is people with advanced HIV disease. In order to clarify it, we modified manuscript title as follow: “Diagnosis of Progressive Disseminated Histoplasmosis in Advanced HIV: a Meta-analysis of Assay Analytical Performance”, and additionally, we described study limitations in the conclusion section as follows: “Finally, this systematic review is limited to the analysis assays performance on PDH in PLHIV, due the variety of histoplasmosis clinical presentations, further systematic reviews and meta-analysis are needed”.

The section dealing with the molecular diagnosis could be improved including a brief revision of the sequences amplified, amplicon sizes, etc.

R. Thanks, we added target genes in the result section.

The addresses of the websites used for literature search should be included in the manuscript.

R. Thanks, more detail about search and meta-analysis are available in the manuscript supplementary material.